# On the Scale Invariance in State of the Art CNNs Trained on ImageNet †

Mara Graziani [1,2,*] , Thomas Lompech [3,‡], Henning Müller [1,2] , Adrien Depeursinge [2,4] and Vincent Andrearczyk [2]

1 Department of Computer Science (CUI), University of Geneva (UNIGE), 1227 Carouge, Switzerland; henning.mueller@hevs.ch
2 Institute for Business Information Systems, University of Applied Sciences Western Switzerland (HES-SO Valais), 3960 Sierre, Valais, Switzerland; adrien.depeursinge@hevs.ch (A.D.); vincent.andrearczyk@hevs.ch (V.A.)
3 Institute National Polytechnique de Toulouse, Ecole Nationale Supérieure d'Electrotechnique, d'Electronique, d'Informatique, d'Hydraulique et des Télécommunications (INP-ENSEEIHT), 31000 Toulouse, France; thomas.lompech@gmail.com
4 Nuclear Medicine and Molecular Imaging Department, Centre Hospitalier Universitaire Vaudois (CHUV), 1011 Lausanne, Vaud, Switzerland
* Correspondence: mara.graziani@hevs.ch
† This paper is an extent version of our published research in the Workshop on Interpretability of Machine Intelligence in Medical Image Computing (iMIMIC) at the International Conference on Medical Image Computing and Computer-Assisted Intervention (MICCAI2020) held in Lima, Peru, on 4–8 October 2020.
‡ Research performed during the internship at the HES-SO.

**Abstract:** The diffused practice of pre-training Convolutional Neural Networks (CNNs) on large natural image datasets such as ImageNet causes the automatic learning of invariance to object scale variations. This, however, can be detrimental in medical imaging, where pixel spacing has a known physical correspondence and size is crucial to the diagnosis, for example, the size of lesions, tumors or cell nuclei. In this paper, we use deep learning interpretability to identify at what intermediate layers such invariance is learned. We train and evaluate different regression models on the PASCAL-VOC (Pattern Analysis, Statistical modeling and ComputAtional Learning-Visual Object Classes) annotated data to (i) separate the effects of the closely related yet different notions of image size and object scale, (ii) quantify the presence of scale information in the CNN in terms of the layer-wise correlation between input scale and feature maps in InceptionV3 and ResNet50, and (iii) develop a pruning strategy that reduces the invariance to object scale of the learned features. Results indicate that scale information peaks at central CNN layers and drops close to the softmax, where the invariance is reached. Our pruning strategy uses this to obtain features that preserve scale information. We show that the pruning significantly improves the performance on medical tasks where scale is a relevant factor, for example for the regression of breast histology image magnification. These results show that the presence of scale information at intermediate layers legitimates transfer learning in applications that require scale covariance rather than invariance and that the performance on these tasks can be improved by pruning off the layers where the invariance is learned. All experiments are performed on publicly available data and the code is available on GitHub.

**Keywords:** scale invariance; deep learning; interpretability; medical imaging

## 1. Introduction

Computer vision algorithms trained on natural images must achieve scale invariance for optimal robustness to viewpoint changes. Multi-scale scale invariant approaches are popular in both image processing (e.g., local descriptors, filter banks, wavelets and pyramid scale space [1]) and in recent deep learning techniques [2–5]. Deep Convolutional Neural Networks (CNNs) [6,7] achieve state-of-the-art performance in object recognition tasks with

scale variations (e.g., ImageNet [8]) by implicitly learning scale invariance even without a pre-defined invariant design [9]. Such invariance, together with other learned features of color, edges and textures [10,11], is transferred to other tasks when pretrained models are used to learn from limited training data [12]. Training from scratch is sometimes a preferred alternative to introduce desired invariances in the learned features [13,14]. Scratch training is adopted by scale covariant [4] and multi-scale designs [15–18].

This work is based on the assumption that the scale invariance implicitly learned from pretraining on ImageNet can be detrimental to the transfer to applications for which scale is a relevant feature. With a controlled viewpoint and known voxel spacing dimensions, scale is informative (and often decisive) in some medical imaging tasks (e.g., size of lesions, tumoral regions or cell nuclei, as illustrated in Figure 1. The other transferred features such as shape, color and texture, however, are beneficial to the medical tasks, in particular for learning from limited data and improving the model accuracy and speed of convergence [10,19–21]. We therefore formulate the hypothesis that a specific design retaining helpful features from pretraining while discarding scale invariance can perform better than both a standard transfer and training from scratch. The experiments in this paper focus on validating this hypothesis by identifying the network layers where the invariance to scale is learned and by proposing a way to isolate and remove this unwanted behavior while maintaining the beneficial impact of transfer.

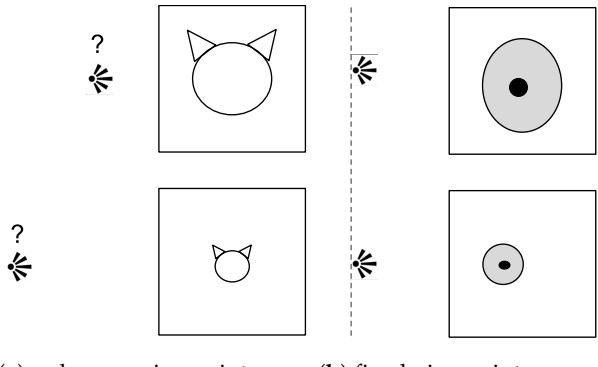

(**a**) unknown viewpoint          (**b**) fixed viewpoint

**Figure 1.** Illustration of (**a**) an unknown and varying viewpoint typical in natural images that requires scale-invariant analysis and (**b**) a controlled viewpoint in which a difference in size carries crucial information that is discarded by a scale invariant analysis.

We make use of deep learning interpretability to preserve the scale covariance of the deep features [22]. The network layers where invariance to scale is learned are identified by applying Regression Concept Vectors (RCVs) [23], a post-hoc interpretability method that uses linear probes [24,25] to determine the presence of a given concept in the network features. This information is used to optimize the transfer by developing a pruning strategy that maintains scale-covariant features without requiring the re-training from scratch in [13,14] or any specific network design. The experiments in this paper extend results, discussions and visualizations of our published research in the Workshop on Interpretability of Machine Intelligence in Medical Image Computing (iMIMIC) at the International Conference on Medical Image Computing and Computer-Assisted Intervention (MICCAI2020) [22] with new in-depth analyses and results. The additional contributions of this paper are stated in the following. New analyses including experiments on image resizing in Section 4.1 and inputs of random noise in Section 4.2 are used to show that object scale and input size have dissociated representations in the CNN layers. While the former is learned from the input data, the latter is shown to be intrinsically captured by the architecture (see Section 4.2). The results on the scale quantification are validated for multiple ImageNet object categories in Section 4.3. The significance of the results on the histopathology task is evaluated by statistical testing in Section 4.4. An additional study

is performed on models trained from scratch for this task, showing that our proposed pruning strategy outperforms both models and pretrained networks in Section 4.4.

The results from this work increase our understanding of scale information in feature reuse. Scale covariance is highest at intermediate layers for all ImageNet object categories, while the invariance is learned in the last dense prediction layer (Section 4.3). This is relevant not only in medical imaging but also in other applications with a controlled viewpoint. Considering this information about scale may help to build models that predict the magnification range of images for which the physical dimension of voxels is unknown, for example, magnification level not reported. For example, remote sensing, defect detection, material recognition and biometrics (e.g., iris and face recognition with registered images) [1]. In the medical context, these results may have a positive impact on the use of large and growing open-access biomedical data repositories such as PubMed Central (https://www.ncbi.nlm.nih.gov/pmc/tools/openftlist/, accessed on 2 April 2021) to extend existing medical datasets [26].

## 2. Related Work

Built-in scale covariance (features that are covariant with a transformation are also referred to as continuous with this transformation.) and invariance in specific CNN designs have been studied and implemented in the literature [2–5,27,28]. While these methods, together with other types of inherent covariance, can alleviate the need for pre-training and large amounts of available data, transfer learning remains extremely common in deep learning applied to medical imaging [21]. As an attempt to understanding CNN behavior with respect to scale, manually selected deep activations were shown to respond to faces viewed at different scales in [29]. Invariance to scale in classic CNN architectures has been analyzed in [30], where the authors use computer-generated images to control attributes (concept measures, including scale) of a single object and visualized the effect on the internal representations. In [9], the regression of geometric image transformations (e.g., image flips and half-rescaling) was studied in an attempt to learn the homomorphic transformations in the feature space that account for the transformations of the input. The authors conclude that scale invariance is implicitly learned on ImageNet as accuracy is not improved by reversing the scaling transformations in the feature space. While [9] learns transformations in the feature space of a trained network, an end-to-end supervised method is proposed in [31] to enforce the disentanglement of transformations including rotations and scales, providing built-in covariance properties. On another line, the vulnerability of CNNs to adversarial attacks with transformations including scaling was studied in [32,33].

Network pruning approaches were proposed in [34,35], with medical applications for PAP smear imaging [36] and Chest X-rays [37]. Pruned networks achieve a similar performance to, if not better [36] than, that of the original network. The asset of network pruning is that even if not providing massive increases in network performance it improves training convergence and it reduces the number of parameters to be trained and thus the computational complexity of the models [37]. This allows the training and fine-tuning of the models on smaller datasets, as shown by the study on PAP smears [36]. Pruning methods mostly focus on identifying the importance of individual elements in the network, such as individual neurons [34], individual filters and/or feature maps [36,37]. Particularly in [35], the authors dealt with multiple object scales by specific-design observations that can make their pruning responsive to multiple object scales. We propose a pruning strategy that, differently from [34,36], focuses on entire layers and that evaluates the layer importance in terms of the scale covariance of the extracted features. Our pruning strategy does not require an explicit design as in [35], nor expensive computations of evolutionary strategies as in [37]. Our method can be applied to any architecture pre-trained on ImageNet inputs to understand the scale covariance of intermediate layers and proposes a pruning strategy that can improve the transfer to applications where object scale is a relevant feature.

Post-hoc interpretability, as defined in the taxonomy of Lipton [38], is particularly suited to the analyses required by this paper since it does not require adding any additional

constraints to the optimization. A post-hoc method can be applied to any model without the need to re-train the parameters. Linear classifier probes [24] were proposed to analyze class-separability at intermediate layers in terms of the classification of the class labels by a linear model. Kim et al. introduced Concept Activation Vectors (CAV) [39] to classify arbitrary concepts (e.g., striped texture) that can be either present or absent in a set of sample images. RCVs [23] extended CAVs to continuous concept measures with a linear regression at intermediate layers of the CNN. This approach led to insightful observations in general computer vision [25,40] and medical imaging [23,41,42].

## 3. Materials and Methods

This section outlines the proposed method and the setups used for the experiments. Section 3.1 introduces the notations in the paper, while Sections 3.2 and 3.3 describe the datasets and network architectures, respectively. We outline the main approach in Section 3.4, while the evaluation metrics are defined in Section 3.5. The hypotheses, scope and methodologies of the multiple experiments are described in Section 3.6.

### 3.1. Notations

We consider an input image $X \in \mathbb{R}^{w \times h}$, where $w$ is the image width and $h$ is the height. The function $\phi(\cdot)$, defined as $\phi : \mathbb{R}^{h \times w} \to \mathbb{R}^d$ maps the input image to a vector of arbitrary dimension $d$. At intermediate layers, the $d$ scalars are obtained from averaged feature maps. At the final fully-connected layer, $\phi(\cdot)$ transforms $X$ into a set of predictions. As further explained in Section 3.4, we analyze the scale information using covariance, defined as the transformation $g' : \mathbb{R}^d \to \mathbb{R}^d$ that predicts the transformation $g : \mathbb{R}^{h \times w} \to \mathbb{R}^{h \times w}$ of the input image $X$ in the feature space obtained by $\phi(g(X))$. The scaling transformations are expressed as $g_\sigma(\cdot)$, being parameterized by a scale factor $\sigma$. We consider images of original size $S_o = h_o \times w_o$ containing a single object that is annotated by a bounding box of size $S_b = h_b \times w_b$. $h_o$ and $w_o$ are respectively the original image width and height. The images for which the size of the object bounding box is approximately equal to the original image size ($S_b \approx S_o$) are referred to as *filled* images.

### 3.2. Datasets

The experiments in this paper involve two datasets since the scale analysis is performed on inputs of natural images and the proposed final architecture is evaluated on a medical image analysis task. For the scale quantification part, images with manual annotations of bounding boxes are selected from the publicly available PASCAL-VOC dataset [43]. We restrict our analysis to three object categories and images containing a single bounding box, chosen among the available annotated classes. These are *albatross* (ID: n02058221, 441 images), *kite* (ID: n01608432, 406 images) and *racing car* (ID: n04037443, 365 images).

For the histopathology application, the data consist of 141 Whole Slide Images (WSI) of Estrogen Receptor-positive Breast Cancer (ERBCa+). For these images of $2000 \times 2000$ pixels, manual annotations of 12,000 nuclei are available [44]. Smaller image regions are extracted as image patches from the WSIs. A total of 69,019 patches with nuclei segmentation masks were split into training, validation and test partitions (approximately 60%, 20%, 20% respectively) as shown in Table 1. To not introduce bias, all the patches from a single image were assigned to the same data partition. The imbalance in the magnification categories is due to the area covered by each magnification level. The average nuclei area is extracted for each input image by computing the average number of pixels in the relative nuclei segmentation mask. Example images with overlaid segmentation masks are displayed in Figure 2.

**Table 1.** Number of Estrogen Receptor-positive Breast Cancer (ERBCa+) patches extracted per magnification and partition.

| Split/# Patches | 5X | 8X | 10X | 15X | 20X | 30X | 40X | Total |
|---|---|---|---|---|---|---|---|---|
| Train | 94 | 2174 | 4141 | 7293 | 9002 | 10,736 | 11,638 | 45,078 |
| Validation | 8 | 588 | 1197 | 2132 | 2604 | 3504 | 3150 | 12,733 |
| Test | 36 | 428 | 900 | 1728 | 2198 | 2802 | 3166 | 11,208 |
| Total | 138 | 3190 | 6238 | 11,153 | 13,804 | 16,592 | 17,904 | 69,019 |

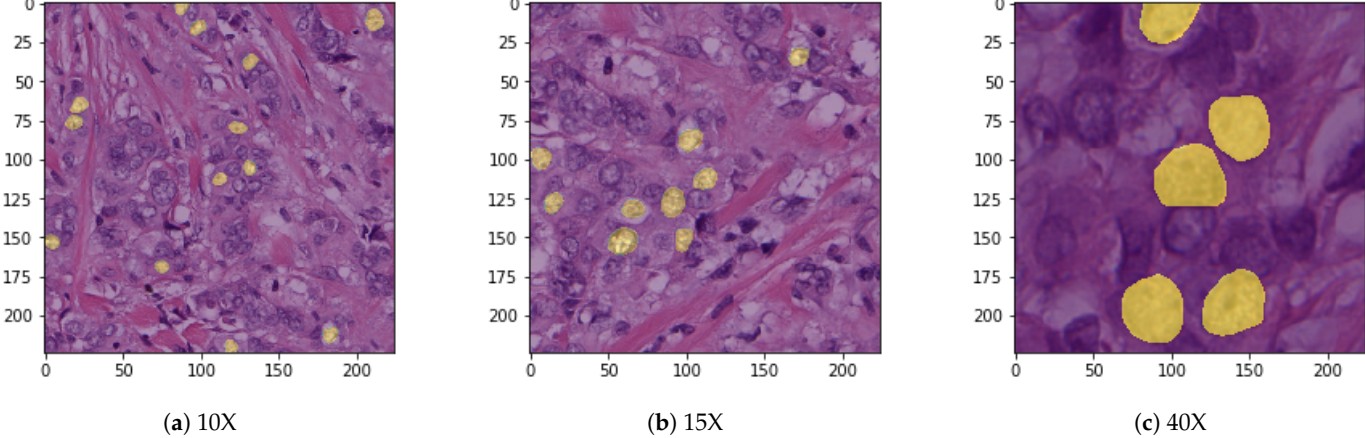

(**a**) 10X  (**b**) 15X  (**c**) 40X

**Figure 2.** Examples of histopathology images at 10, 15 and 40X with nuclei segmentations.

### 3.3. Network Architectures

InceptionV3 [6] and ResNet50 [7] are used for the analysis with pre-trained ImageNet weights. The networks produce a vector of probabilities $f(X) \in [0,1]^{1000}$, where $\sum_{i=1}^{1000} f(X)[i] = 1$. Transfer to the histopathology data is performed from both the original and pruned architectures. To predict the average nucleus area, a single-unit dense layer is trained to minimize the mean squared error loss between the true areas and the predicted ones. The nuclei area is expressed for each image as the average number of pixels within the segmentation of the nuclei present in the image. The magnification category is also obtained from the average nuclei areas. The predicted areas are mapped to the magnification category that has the closest mean average value of the nuclei areas in the training set. This mapping approach was used since it outperforms the direct classification of the magnification in [17]. The networks are implemented in Keras and trained for five epochs with an Adam optimizer and standard hyperparameters (learning rate $1 \times 10^{-4}$, batch size 32 and default values of the exponential decay rates). The full pipeline is shown in Figure 3 and the source code is available on github for reproducibility (https://github.com/medgift/scale_covariant_pruning) (acessed on 2 April 2021).

### 3.4. Quantification of the Scale and Pruning Strategy

Our method quantifies object scale in the input and in the representation space. The act of scaling is defined in image processing as a transformation $g_\sigma(\cdot)$ that generates a new image with a larger or smaller number of pixels, depending on the scaling factor $\sigma$.

In the input space, one may intuitively think of $g_\sigma(\cdot)$ as a reshaping operation. This transformation, however, causes the "train-test" resolution discrepancy in [45] during network inference. We focus this work on images containing a single object, for which we can define image scale as the solid angle of the object in the image, namely the proportion of the field of view occupied by the object [46]. Since a small bounding box corresponds to a smaller space in the field of view of the camera and thus a smaller solid angle, we measure scale in function of the bounding box area $S_b$. Scale measures are thus defined as the ratio $r = \frac{S_b}{S_o} = \frac{h_b \times w_b}{h_o \times w_o}$, where $h_b$ and $w_b$ are the bounding box height and width.

Figure 3 (left) shows two examples of scale measures on input images from the same class appearing at different scales.

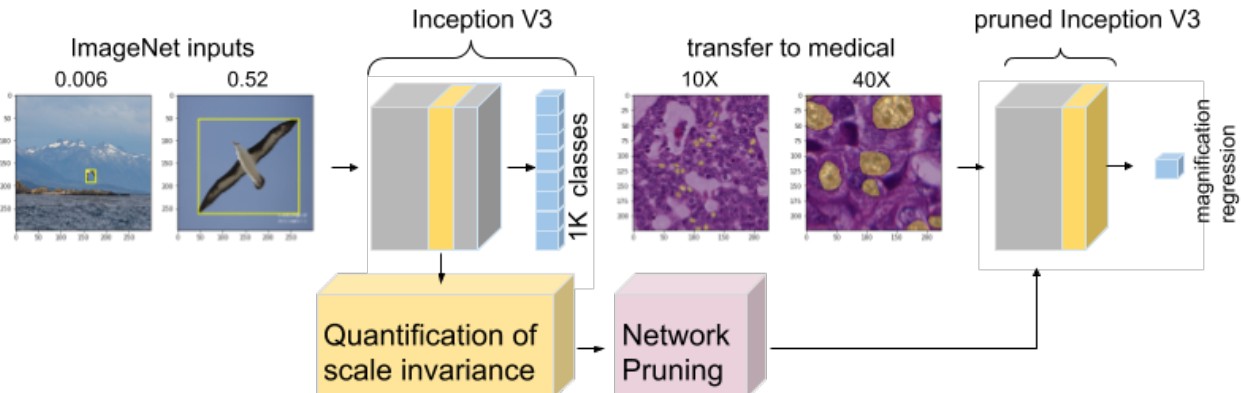

**Figure 3.** Pipeline of scale quantification in ImageNet pretrained networks and consequent network pruning for better transfer to the medical domain. The bounding boxes for the image of the ImageNet class *albatross* and the segmentation masks for the ERBCa+ inputs are overlaid in yellow on the images. The bounding box ratios *r* are reported on top of the ImageNet inputs. ERBCa+ images are shown at magnifications 10X and 40X. The layer evidenced in yellow is the most informative about scale according to our quantification of scale invariance. The pruned network drops the layers after this for solving the medical task. Best seen on screen.

In the feature space, we aim at finding a linear transformation $g'_\sigma(\cdot)$ that is a predictable transformation of $g_\sigma(\cdot)$ in the input space. We start by using the definitions of invariance (1) and covariance (2) of a mapping $\phi(\cdot)$ to a transformation $g(\cdot)$ as follows

$$\phi(g(\cdot)) = \phi(\cdot), \tag{1}$$

$$\phi(g(\cdot)) = g'(\phi(\cdot)). \tag{2}$$

In our analysis, we consider functions of the input image X. We evaluate the covariance of the function $\phi(X)$, that is, whether we can find a transformation $g' : \mathbb{R}^d \to \mathbb{R}^d$ in the feature space that predicts a transformation $g : \mathbb{R}^{h \times w} \to \mathbb{R}^{h \times w}$ of the input image (In these terms, equivariance is a particular case of covariance, when $g'(\cdot) = g(\cdot)$. The equivariance implies that the function $\phi(\cdot)$ maps an input image to a function in the same domain, not relevant in our scenario). $\phi(X)$ consists of $d$ scalars representing either the averaged feature maps of intermediate layers or the activations of fully-connected layers. To find the transformation $g'_\sigma(\cdot)$, we search a regression vector **v** (i.e., the RCV [23]) in the feature space to predict the scaling factor $\sigma$ as (For simplicity, we omit the intercept. In Equation (3), the intercept is $v_0$ with $\phi_0(g_\sigma(X)) = 1$.):

$$\sigma = \sum_i v_i \phi_i(g_\sigma(X)) = \mathbf{v} \cdot \phi(g_\sigma(X)). \tag{3}$$

We then have that $g'_\sigma(\cdot)$ can be represented as a translation matrix (in $\mathbb{R}^d$) by $\sigma$ along **v**, so that $g'_\sigma(\phi(X)) = \phi(X) + \mathbf{v} \cdot \sigma$. The proposed pruning strategy compares the test $R^2$ (determination coefficient) of the regression vectors obtained at multiple depths to identify the layer where the scale covariance is the highest. The layer with the highest test $R^2$ (the yellow layer in Figure 3) is where the scale covariance is the highest. Layers deeper than this one are pruned off the architecture and a GAP operation is added to obtain a vector of aggregated features.

*3.5. Evaluation*

In Section 4.1, the network performance is monitored in terms of top-5 accuracy and average probability of the correct class for a set of $N$ inputs, that is, $p = \frac{1}{N} \sum_{j=1}^{N} f(X_j)[y_j]$

for ground-truth labels $y_j$. The regression of image size in Section 4.2 and the RCV of scale in Section 4.3 are evaluated on held-out images using the $R^2$ determination coefficient (We compute $R^2 = 1 - \frac{\sum_{i=1}^{N}(s_i - \hat{s}_i)^2}{\sum_{i=1}^{N}(s_i - \bar{s})^2}$, were $N$ is the number of test data samples, $\hat{s}_i$ is the size predicted by the regression model, $\bar{s}$ is the mean of the true sizes $\{s_i\}_{i=1}^{N}$. A similar formulation applies to the evaluation of the regression of the scale ratio $r$). To keep the test $R^2$ within a [0,1] range for visualization and comparison, we report the normalized test $R^2 = \frac{e^{R^2}}{e} = e^{R^2 - 1}$, for which values below $\frac{1}{e} \approx 0.37$ evidence bad performance.

The transfer learning experiments on the histopathology task in Section 4.4 are evaluated by the Mean Average Error (MAE) and Cohen's kappa coefficient. MAE is used to evaluate the regression of the average nuclei areas, while Cohen's kappa coefficient is used to measure the inter-rater reliability of the prediction of the magnification classes.

### 3.6. Experimental Setups

In the following, we clarify the objectives and setups of the performed experiments. The experiments in Section 4.1 address the main hypothesis that CNNs pretrained on ImageNet are invariant to transformations of the input size. We want to show, in particular, that this behavior is also true for images containing objects that naturally appear at various scales due to varying viewpoints (for which an example is given in Figure 4). To show this, we set up two related experiments. In the first experiment, we use *filled* images containing only one object covering the entire space in the image (i.e., $S_b \approx S_o$), which are selected manually from the pool of ImageNet validation images. In the experiment, each *filled* image is reshaped to a squared input of arbitrary size and the network output is monitored by checking the probability of the correct class (top-1 accuracy) and the top-5 accuracy. We use the Lanczos interpolation (Similar results were obtained using bilinear, nearest, bicubic and Lanczos interpolations.) to reshape the images to a squared input of $S_i = s_i \times s_i$, with $s_i$ ranging from 75 to 500 pixels. A total of 69 images of size $S_o = 500 \times 500$ and other 69 of smaller original size (mean $\bar{S}_o = 285 \times 285$) were used. In the first set, images are either reduced or increased in size by the interpolation, whereas in the latter they are only reduced. In the second experiment, we separate the impact of input size $S_i$ from that of object size $S_b$. We do not use *filled* images anymore and we release the condition $S_b \approx S_o$. In other words, we compare images that are resized to the same sizes $S_i$, but that contain objects of different sizes.

Section 4.2 further analyses the difference between changing input size and object scale. We formulate the main hypothesis that the scaling operation $g_\sigma(\cdot)$ cannot be performed as a simple input reshaping operation because the CNN features encode information about image size differently from object scale. We hypothesize that information about image size is encoded in the features from the padding effect of early convolutional layers. To verify this, we introduce the corrected Global Average Pooling (GAP) illustrated in Figure 5. This operation averages only the activations of the neurons with a receptive field contained entirely in the input image. This is in practice equivalent to discarding the activations at the border of the feature maps that are affected by padding operations. Images of white noise of different sizes are used for this experiment, since they do not contain any object nor related scale. These images are generated by sampling pixel values from a uniform distribution in the range $[0, 255]$. The experiment aims at regressing the image size for these noise inputs in the intermediate layers of the CNN. If the network encodes information about the image size differently from object scale, then we should be able to regress the size from the noise inputs. If this information is encoded from the padding at early layers, then the regression with the corrected GAP should fail as this operation discards the edges of the feature maps. We thus compare the regression of image scale with and without the corrected GAP to show that current state-of-the-art CNN architectures encode information about the image size. The regression vector **v** in Equation (3) is sought to regress the image width $s_i$. Since the receptive fields grow throughout the network, the region of activations unimpacted by the paddings reduces up to a point where no activation remains

for the corrected GAP. Because of this limitation, we can only use this method to show the impact of zero-padding but we cannot use it for the analysis of scale invariance throughout the network.

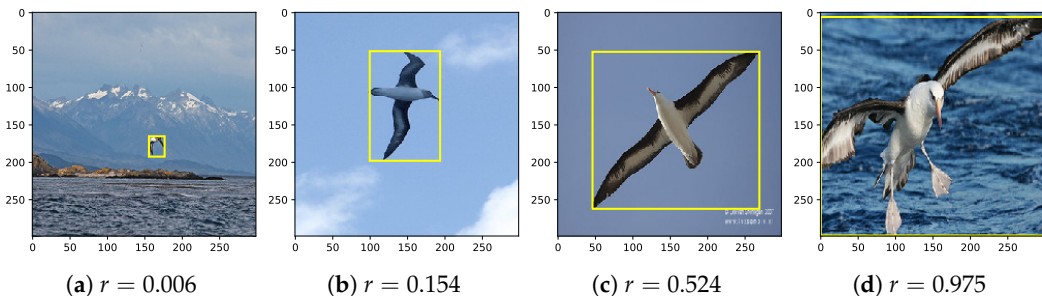

(**a**) $r = 0.006$  (**b**) $r = 0.154$  (**c**) $r = 0.524$  (**d**) $r = 0.975$

**Figure 4.** Examples of albatross images and their respective scale concept measures $r = \frac{S_b}{S_o}$ used for learning the regression.

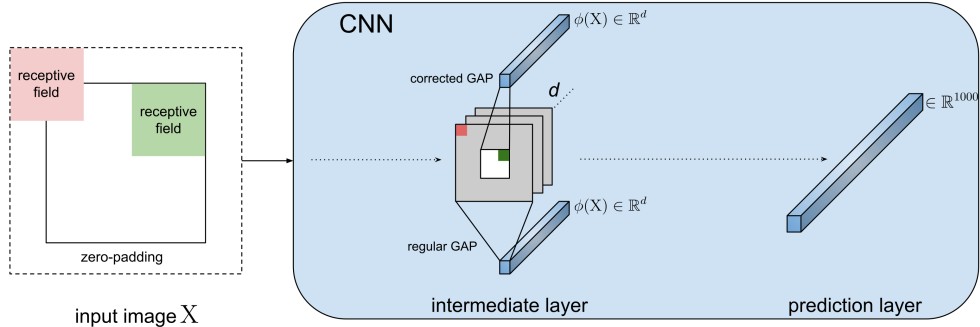

**Figure 5.** Illustration of the working principle of the corrected Global Average Pooling . The colored receptive fields in the input image (**left**) are associated with the colored neurons in the feature maps (**center**). In the Convolutional Neural Network (CNN), activations used for the corrected GAP (**top**) are displayed in white that is, activations of the neurons with a receptive field contained in the input image. All activations are used for the regular GAP (**bottom**). Best viewed in color.

In the next experiments, we use images with fixed input size to $S_i = 299 \times 299$ because of our hypothesis that input size and object scale are learned in different ways. The measures of scale are based on the ratio $r$ defined in Section 3.4. The experiments in Section 4.3 focus on the regression of scale measures in ImageNet pretrained models for the object categories albatross (ID: n02058221), race car (ID: n04037443) and kite (ID: n01608432).

We use 70% of the input class images to learn the regression, while the remaining images are held out for evaluating the determination coefficient. Examples of images and their corresponding scale concept measures $r$ are shown in Figure 4 for the albatross class. Finally, we run experiments on the transfer to the histopathology task in Section 4.4. The information extracted in the previous experiments is used to improve the transfer of pretrained features to the medical imaging task in [17]. This is obtained by implementing the pruning pipeline summarized in Figure 3. The medical task in these experiments is the regression of the average area of the nuclei in histopathology images. The pruning of network layers is performed by comparing the test $R^2$ on the natural images (Figures 8a and 9 for InceptionV3 and Figure 8b for ResNet50) to identify the layer where the scale covariance is the highest. This evaluation is averaged across object categories to remove the dependence on the class of the inputs (see Appendix A.1, Figure A2).

## 4. Results

### 4.1. Invariance of the Predictions to Resizing

This section contains the results of the first two experiments described in Section 3.6. The CNN predictions for reshaping transformations of the *filled* images are reported in Figure 6. The two subsets of images being used do not report marked differences.

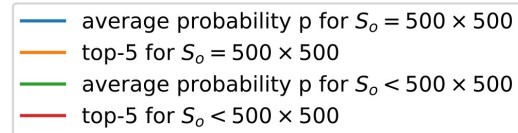

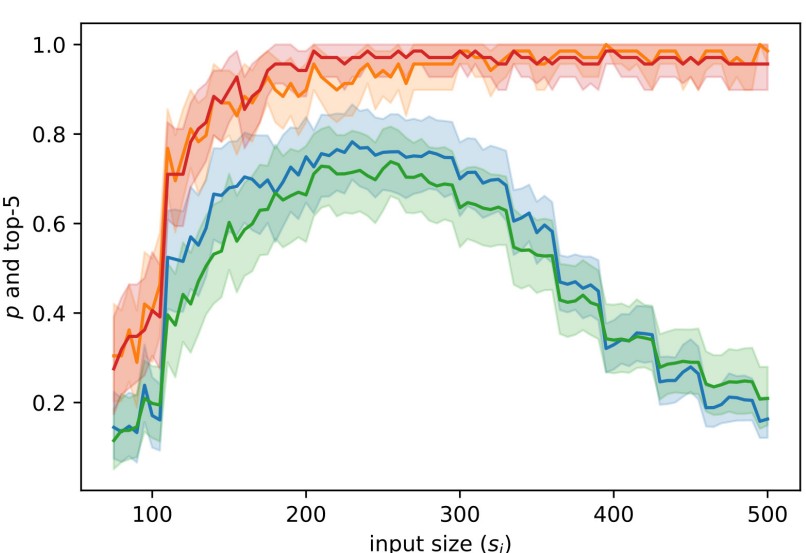

**Figure 6.** Average probability $p$ of correct class and top-5 accuracy vs. input size $s_i$ for 69 *filled* images with $S_o = 500 \times 500$ and 69 *filled* images with $S_o < 500 \times 500$ ($\bar{S}_o = 285 \times 285$). The 95% confidence intervals are reported.

The results of the second experiment are not reported for brevity and because they are very similar to those in Figure 6. The predictions resulted in only slightly better results with *filled* images ($S_b \approx S_o$) and we did not notice a shift of lower probabilities towards smaller input sizes when the objects are smaller ($S_b < S_o$). This shows that $S_i$ is more relevant for the predictions than the object size $S_b$, as expected.

### 4.2. Experiments on Noise Inputs

In this section, we use input of white noise that does not contain any object nor scale information. The regression of $s_i$ is learned from five noise images (We intentionally use a small number of images to illustrate the simple linear correlation. Similar results are obtained when using more images.) and evaluated on 20 held-out images.

The results show that we can regress the size for the model with the regular GAP in deep layers, with the $R^2$ close to one in Figure 7a. On the contrary, Figure 7b shows that we cannot regress the size information when aggregating the feature maps using the corrected GAP ($R^2 < 0$).

In light of these results and those in Section 4.1, we do not associate the input size to the measure of object scale in the analyses of the next section.

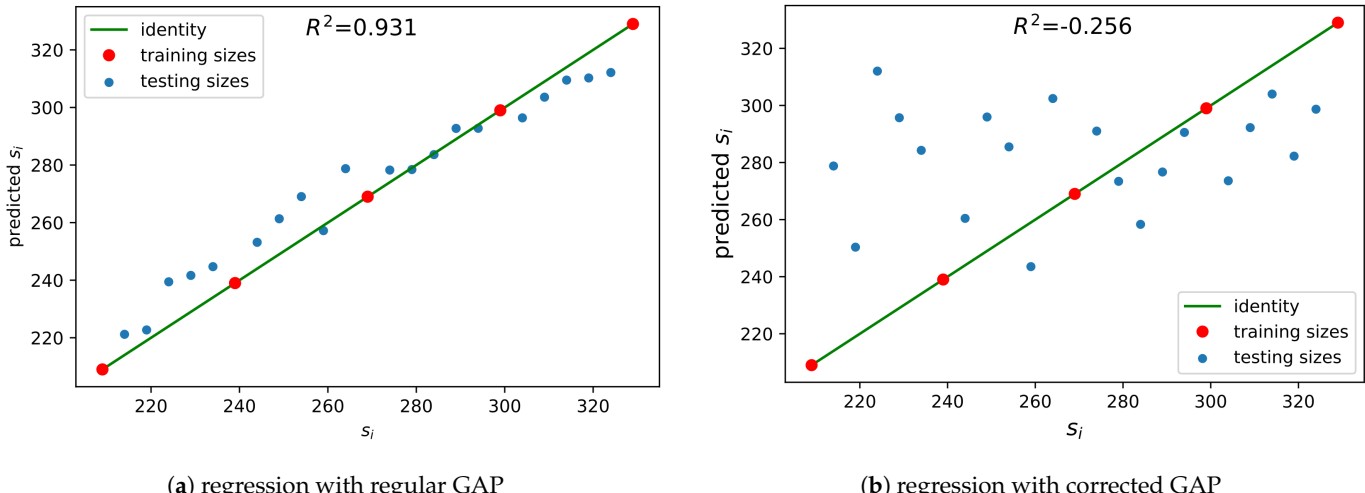

(**a**) regression with regular GAP  (**b**) regression with corrected GAP

**Figure 7.** Regression of size $s_i$ with noise inputs: (**a**) Regression at layer *mixed0* for regular GAP; (**b**) Regression at layer *mixed0* for corrected GAP. The regression is evaluated as the $R^2$ of the prediction of scale measures on held-out noise images.

### 4.3. Layer-wise Quantification of Scale Covariance

In this section, we start by focusing on 441 images of the albatross class containing a single object bounding box. Later in the section, the experiments are extended to the race car and kite classes.

In Figure 8a, we compare the scale regression in a randomly initialized InceptionV3 with one trained on ImageNet. We regress the scale concept measures in activations at different depths, as explained in Section 3.4. We also compare to a baseline in which the regression is trained with random concept measures, that is, shuffling the scale concept measures before regression. As explained in Section 3.5, we report the fraction $\frac{e^{R^2}}{e}$ to visualize positive values in the presence of large variations in negative values (as low as −11,077). The detailed values of $R^2$ are reported in Table A1 of the Appendix A.2. Values of $R^2$ close to one reflect the linear covariance of the intermediate layers to object scale as defined in Section 3.4. Values below zero reflect the invariance to scale.

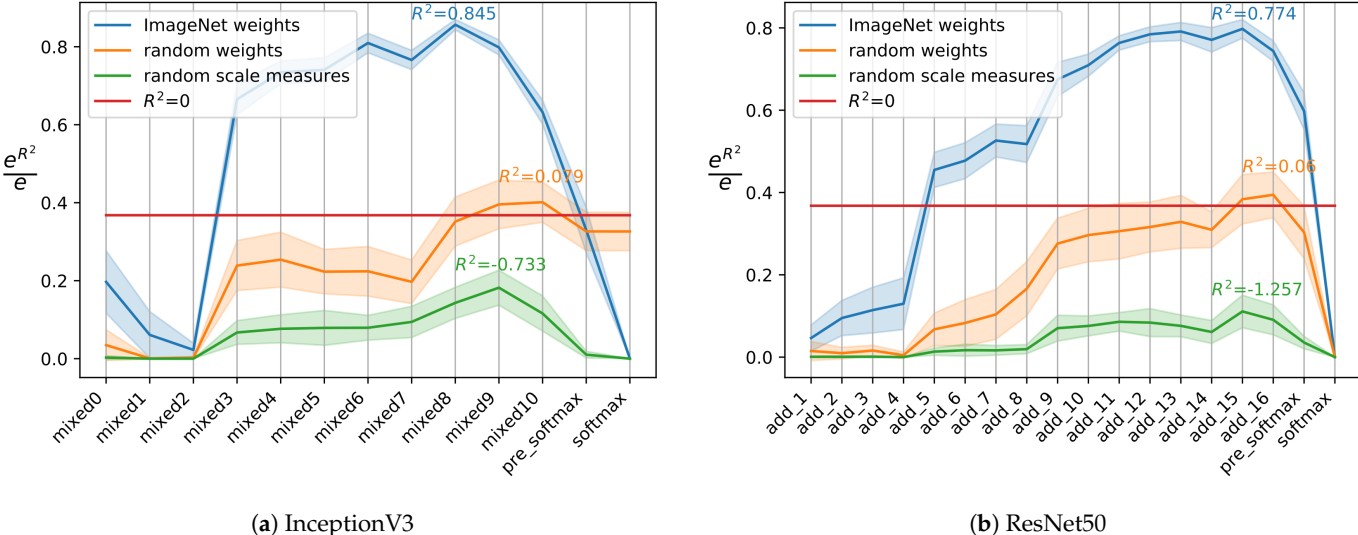

(**a**) InceptionV3  (**b**) ResNet50

**Figure 8.** Comparison of regression (RCV) of scale measures at different layers on the albatross ImageNet class (ID: n02058221). The regression is evaluated as the $R^2$ of the prediction of scale measures on held-out images and $\frac{e^{R^2}}{e}$ is plotted for better visualization. Values above the red line $R^2 = 0$ show a predictive regression better than the average of ratios $r$. Average and standard deviations are reported for 25 runs.

The results are similar on the other two classes, for which the results in InceptionV3 are reported in Figure 9. The results on the ResNet50 architecture are also similar and can be seen in Figure 8b and in Figure A1 of the Appendix A.1.

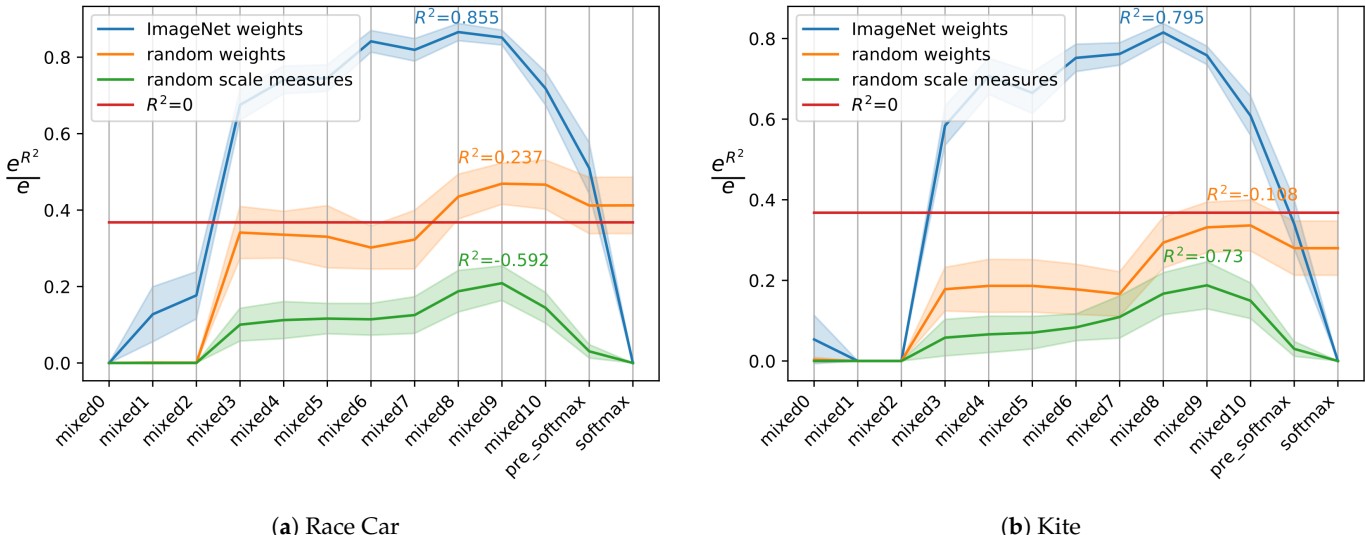

(**a**) Race Car                                                                 (**b**) Kite

**Figure 9.** Comparison of regression (RCV) of scale measures at different InceptionV3 layers on the two classes (**a**) race car (ID: n04037443) and (**b**) kite (ID: n01608432). The regression is evaluated as the $\frac{e^{R^2}}{e}$ of held-out images. Values above the red line $R^2 = 0$ show a predictive regression better than the average of the scale ratios $r$. Average and standard deviations are reported for 25 runs.

### 4.4. Improvement of Transfer to Histopathology

The original InceptionV3 and ResNet50 networks are compared to their pruned counterparts in terms of performance in the nuclei area and magnification prediction in Table 2. We report the Mean Average Error (MAE) across ten repetitions (different seeds were used to initialize the dense connections to the last prediction layer.)and the relative standard deviation for the prediction of the average area. We also report Cohen's kappa coefficient for the prediction of the magnification category. For both evaluated networks and both tasks (area and magnification prediction) the results show significant improvements when the networks are pruned at the relevant layer, validating the proposed scale invariance analysis in the previous sections. The non-parametric Wilcoxon signed-rank test was used to evaluate the statistical significance ($p$-value < 0.001 for the MAE and kappa with both networks). The average MAE (standard deviations reported in brackets) between the true nuclei areas and those predicted by the pruned Inception V3 are respectively 55.33 (31.16) for 5X images, 42.15 (11.39) for 8X, 34.65 (0.15) for 10X, 33.28 (0.69) for 15X, 48.38 (5.26) for 20X and 81.05 (15.67) for 40X images.

**Table 2.** Mean Average Error (MAE) of the nuclei area regression (in pixels) and Cohen's kappa coefficient between the true and predicted magnification categories. Results are averaged across ten repetitions; the standard deviation is reported in brackets.

| Model | Layer | MAE (std) |
|---|---|---|
| pretrained IV3 | mixed10 | 81.85 (11.08) |
| from scratch IV3 | mixed10 | 82.30 (17.92) |
| pruned IV3 | mixed8 | **54.93**  (4.32) |
| pretrained ResNet50 | add16 | 70.08 (12.49) |
| from scratch ResNet50 | add16 | 95.66 (21.39) |
| pruned ResNet50 | add15 | **54.76** (3.10) |

## 5. Discussion

In this section, we discuss the results and give further insights regarding their interpretation, referring to previous studies in the field that support our hypotheses.

We first analyze the relationship between image size and object scale. Our first experiment in Section 4.1 reported in Figure 6 shows that the average probability for the correct class is approximately invariant to input sizes in the range $[175, 300]$. Images in this range are likely seen during training since 299 is the size used to train the classification task on ImageNet. The top-5 accuracy is the maximum and unlike $p$ plateaus for $s_i > 200$. This is explained by probabilities being more spread across classes, yet highest probabilities are still given to the correct classes. The invariance of the predictions to the upsizing or downsizing of the original image size, also discussed in this section, confirms that the interpolation used for down and up-sampling has a neglectable influence on the predictions (bilinear, nearest, bicubic or Lanczos). The experiment with resized images containing objects of multiple sizes shows that the information of input size prevails on the one of scale when these two are not correctly separated. A similar yet less detailed analysis performed in [45] showed an increase of top-1 accuracy when training and testing sizes approximately match. The strong encoding of information about the input size within the network is attributed by the authors in [45] to the change in the distributions of the ReLU activations of deep layers for smaller input images. We further support our analysis with the experiments on noise inputs in Section 4.2. The white noise images do not contain any object and the information about image size is captured also in these images (as shown by the results in Figure 7a). By introducing the corrected GAP, we show that the regression of image scale in noise images is mostly due to the padding effects at early convolution layers that encode information about the input size. In Figure 7b, we confirm this hypothesis by showing the poor performance of the linear regression when removing the information on input size by manually correcting the GAP.

From the quantification of scale covariance in Section 4.3, we observe that information about scale is present at intermediate layers, and that invariance is reached only towards the last layers before softmax. Comparing the regression in the intermediate CNN layers of real concept measures (reported in blue in Figure 8a,b) and those of random concept measures (reported in green in the same figures), we conclude that the scale information is present at intermediate layers. We can linearly regress the true scale ratios better than random values of scale, with $R^2$ close to one. The $R^2$ of the randomly initialized model weights are close to the ones obtained with random concept measures and less than zero for almost all layers. This shows that an architecture with random weights does not contain information of scale and that this information is learned during network training. The low $R^2$ in the early layers of the trained networks seems to be due to the size of the receptive field, which is too narrow for correctly regressing the input scale. This was also supported by the previous results in [47,48], which suggest that early layers focus on local textures and small object parts. We show this further in the Appendix A in Figure A3, by visualizing the internal features at different depths. Primitive features of color and texture are not sufficient for regressing the object scale. The more complex features of object parts learned after the *mixed2* layer, enable this regression. Finally, the drop in the regression prediction at the end of the trained network shows that scale invariance is learned in deep layers, mostly in the last dense layer (pre- and post-softmax).

The task analyzed in the final experiment for improving the transfer of the learned weights to histopathology data represents an important problem in this field. Many open access repositories (e.g., PubMed Central) do not provide information about the magnification level of the images, which become thus difficult to integrate with other datasets. Data from open access repositories or social networks can provide examples of rare and under-represented cases since these images are often presented for visual comparison and discussion among experts [26]. The proposed pruning strategy drops the layers with scale-invariant features to improve the transfer and better regress the magnification level of histopathology images. For InceptionV3, the pruned features are

a result of a GAP on top of the *mixed8* features. As shown in Table 2, the MAE = 54.93 of the nuclei area regression in *mixed8* is significantly lower than the MAE = 81.85 in *mixed10*. This corresponds to a better prediction of the magnification range, hence to a higher kappa coefficient. The pruned architectures provide a reduction in complexity, requiring the training of 51% and 19% less of parameters respectively for InceptionV3 and ResNet50.

## 6. Conclusions

In this paper, we designed and used an experimental approach to analyze the covariance to object scale in CNNs trained on ImageNet. We then used the analysis of state-of-art CNNs to improve the transfer of these pre-trained networks on a medical task. We made the main distinction between input size and object scale, showing that these two measures should be properly separated to interpret the scale covariance of CNN features. Our scale quantification with the regression of scale ratios represents an intuitive and easy-to-apply method to determine the invariance to scale of intermediate network features. We showed that deep features (up to the penultimate layer) are linearly scale-covariant. These pre-trained features can therefore safely be used either as feature extractor or fine-tuning for tasks in which the scale provides crucial information.

Our network pruning strategy can improve transfer by maintaining the scale-covariance of the features without requiring any explicit design or retraining of the network weights and can thus be applied to state-of-the-art CNNs pre-trained on ImageNet. The proposed pruning largely improves the prediction of magnification in histopathology images.

We recognize the limitations of the proposed work, including the linearity of the regression, where information about scale can be present but impossible to regress linearly. In future work, we will investigate non-linear regression and manifold learning of the feature space.

**Author Contributions:** Conceptualization, M.G., A.D. and V.A.; methodology, M.G.; software, M.G. and T.L.; validation, A.D., V.A. and H.M.; formal analysis, M.G. and V.A.; investigation, M.G.; resources, H.M.; writing—original draft preparation, M.G.; writing—review and editing, M.G., A.D. and V.A.; visualization, T.L. and V.A.; supervision, V.A. and H.M.; project administration, H.M.; funding acquisition, H.M. All authors have read and agreed to the published version of the manuscript.

**Funding:** This research was funded by the Swiss National Science Foundation grant number 205320_179069 and and the European Union's Horizon 2020 project PROCESS grant number 777533.

**Data Availability Statement:** The data used for this study are publicly available and can be downloaded at www.image-net.org and andrewjanowczyk.com/deep-learning/ (accessed on 2 April 2021).

**Acknowledgments:** We would kindly acknowledge the support of Sebastian Otálora concerning the task of magnification regression for histopathology images.

**Conflicts of Interest:** The authors declare no conflict of interest.

## Abbreviations

The following abbreviations are used in this manuscript:

CNN    Convolutional Neural Networks
RCVs   Regression Concept Vectors
CAV    Concept Activation Vectors

## Appendix A

*Appendix A.1. Extended Regression Results*

We report extended results on the scale regression experiments presented in Section 4.1. In Figure A1, we report results similar to Figures 8 and 9 obtained with ResNet50 for the classes race car (365 images) and kite (406 images), not reported in the

main paper for brevity. In each class, 70% of the images are used for learning the regression, the remaining 30% are used to evaluate it.

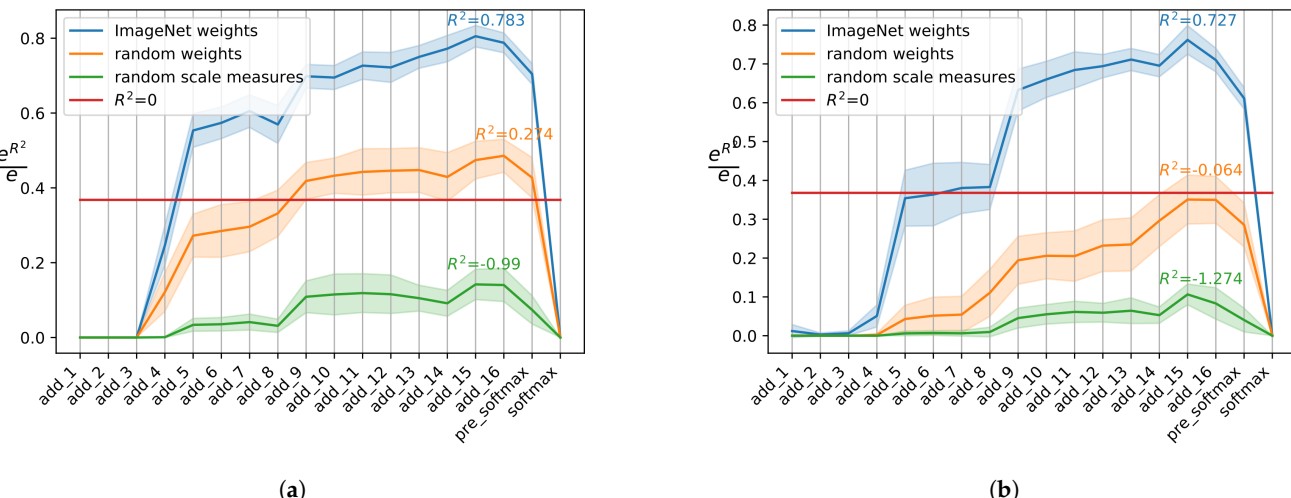

(**a**)                                                                                (**b**)

**Figure A1.** Comparison of regression (RCV) of scale measures at different ResNet50 layers on the classes: (**a**) race car (ID: n04037443), (**b**) kite (ID: n01608432). The regression is evaluated as $\frac{e^{R^2}}{e}$ on held-out images.

In Figure A2, we report the results, averaged across classes, that were used to select the pruning layer for both architectures. As mentioned in Section 4.4, we remove the dependency of the evaluation on the image selection (by using multiple splits) and category (by analyzing multiple classes). We average the results across 10 repetitions for all classes, with a total of 30 evaluations. The evaluation was performed for ten splits of images.

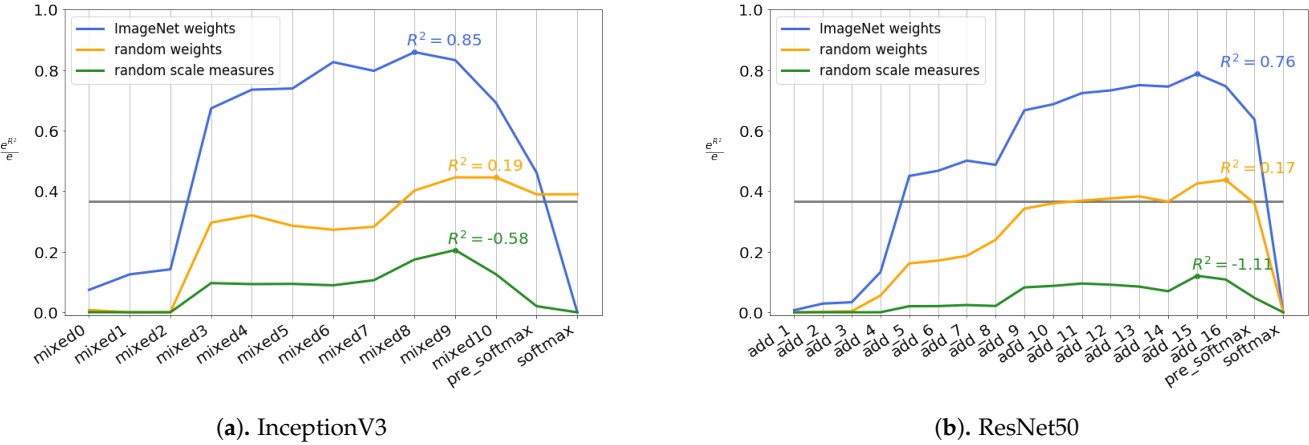

(**a**). InceptionV3                                                        (**b**). ResNet50

**Figure A2.** Average performance (for all classes) of the regression of scale measures on test data at different layers.

The difference in softmax regression between randomly initialized InceptionV3 and ResNet50 for all classes is notable. This can be explained by the softmax probabilities of InceptionV3 being uniformly distributed around $\frac{1}{1000}$ for all 1000 classes as opposed to the sparsely high probabilities of ResNet50. This difference in the probability distribution is due to the different pixel values normalization used in the input pre-processing of the networks. Interesting preliminary analyses of the probability distributions further support these claims, yet this is out of the scope of this paper and will be analyzed in future work.

*Appendix A.2. Detailed Determination Coefficients*

In Table A1, we report the values of $R^2$ obtained on the regression evaluation of the

scale measure at different layers of InceptionV3 with images of the albatross class. The $R^2$ values were plotted in Figure 8a as $\frac{e^{R^2}}{e}$ due to their range.

**Table A1.** Details of $R^2$ of scale measure displayed in Figure 8a. Average of the $R^2$ and standard deviation across 25 runs with InceptionV3 on the albatross class.

| Model | InceptionV3 ImageNet | InceptionV3 Random Weights | InceptionV3 Random Scale Measures |
|---|---|---|---|
| mixed0 | $-0.72 \pm 0.47$ | $-283 \pm 1365$ | $-5.75 \pm 1.38$ |
| mixed1 | $-2.39 \pm 1.21$ | $-331 \pm 1082$ | $-20.1 \pm 6.96$ |
| mixed2 | $-3.12 \pm 0.88$ | $-14.4 \pm 26.1$ | $-20.6 \pm 11.0$ |
| mixed3 | $0.59 \pm 0.06$ | $-0.47 \pm 0.30$ | $-1.81 \pm 0.48$ |
| mixed4 | $0.69 \pm 0.04$ | $-0.41 \pm 0.31$ | $-1.66 \pm 0.42$ |
| mixed5 | $0.70 \pm 0.04$ | $-0.53 \pm 0.26$ | $-1.68 \pm 0.52$ |
| mixed6 | $0.79 \pm 0.03$ | $-0.54 \pm 0.29$ | $-1.62 \pm 0.41$ |
| mixed7 | $0.73 \pm 0.03$ | $-0.66 \pm 0.29$ | $-1.45 \pm 0.43$ |
| mixed8 | $0.84 \pm 0.01$ | $-0.06 \pm 0.19$ | $-0.98 \pm 0.27$ |
| mixed9 | $0.77 \pm 0.02$ | $0.06 \pm 0.16$ | $-0.73 \pm 0.24$ |
| mixed10 | $0.54 \pm 0.05$ | $0.08 \pm 0.13$ | $-1.24 \pm 0.45$ |
| pre-soft. | $-0.12 \pm 0.20$ | $-0.13 \pm 0.15$ | $-3.90 \pm 0.86$ |
| softmax | $-3861 \pm 5729$ | $-0.13 \pm 0.15$ | $-11{,}077 \pm 26{,}421$ |

*Appendix A.3. Visualization of Early Layer Features*

As mentioned in the Discussion, early layers focus mostly on local pixel neighborhoods, not extracting sufficient information to regress the scale ratio in Figures 8a and 9. To support this claim, we use the Lucid toolbox (https://github.com/tensorflow/lucid, accessed on 2 April 2020) to visualize the internal features of InceptionV3 at different depths. As shown in Figure A3, early layers in InceptionV3 mostly focus on simple patterns and colors (see Figure A3a,b). Only at deeper layers it is possible to recognize object parts as in Figure A3c and entire dog faces in Figure A3d.

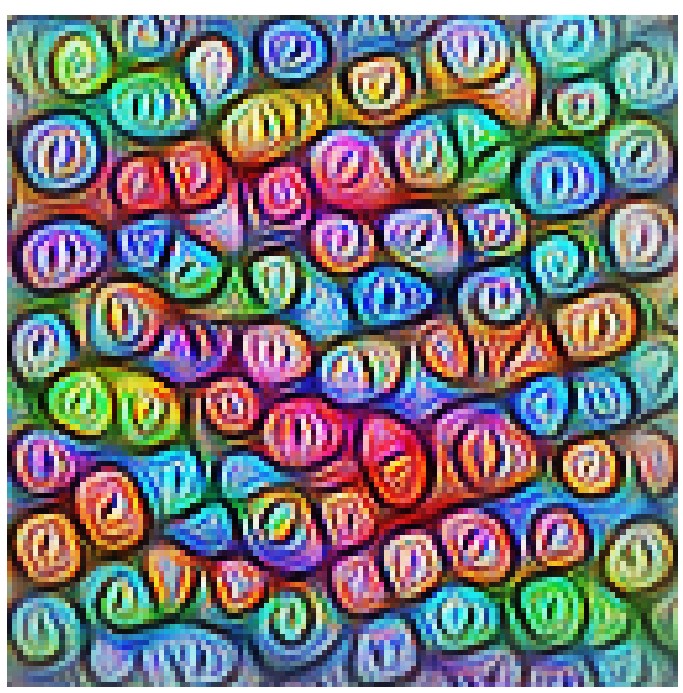

(**a**) Colored pattern from mixed_0.

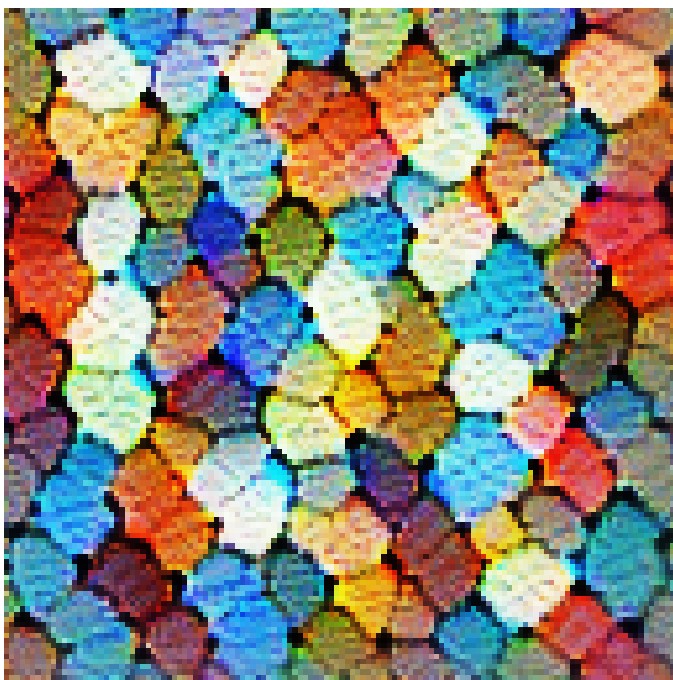

(**b**) Colored pattern from mixed_1.

**Figure A3.** *Cont.*

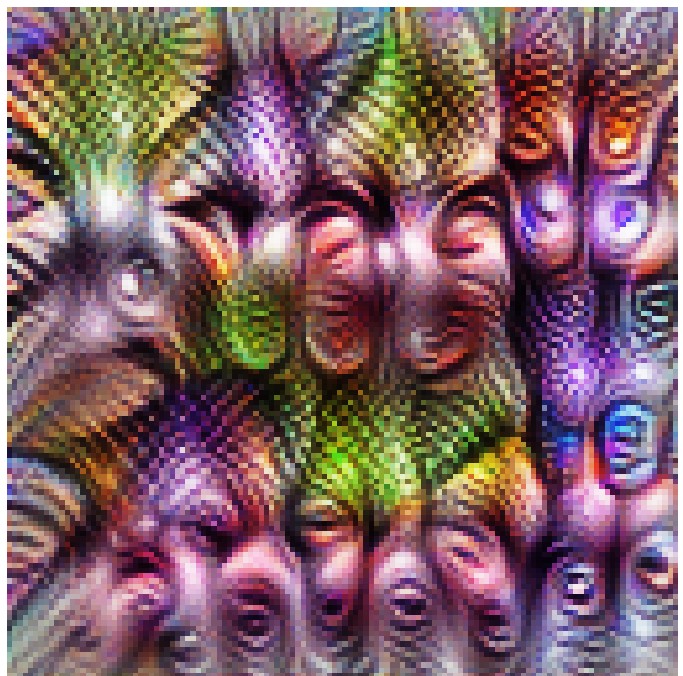 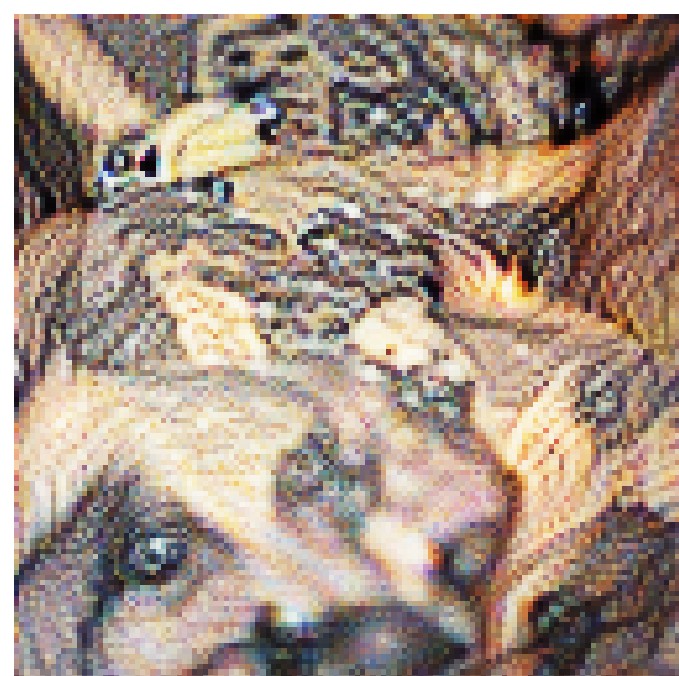

(**c**) Parts of faces and eyes in mixed_2.    (**d**) Dog faces in mixed_6

**Figure A3.** Internal visualization of InceptionV3 unit activations at multiple layer depths, obtained with the Lucid toolbox.

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
