# Peer review of "On the Scale Invariance in State of the Art CNNs Trained on ImageNet†"

_make, doi:10.3390/make3020019_

Round 1
Reviewer 1 Report
The authors studied the scale invariance of the state-of-the-art CNNs pretrained on ImageNet, and found that scale information peaks at central CNN layers and drops close to softmax, where invariance is reached. Such information is further utilized to enhance the performance of medical tasks. I enjoyed reading this paper and would like to suggest accepting this paper for potential publication after minor corrections.
L269: "Fig 3" should be "Figure 3".
In Figure A3, some texts were not shown correctly.
Author Response
Point 1: I enjoyed reading this paper and would like to suggest accepting this paper for potential publication after minor corrections. L269: "Fig 3" should be "Figure 3".
Response 1: Dear Reviewer, thank you for such positive feedback and for the useful suggestions. The reference to Figure 3 has now been corrected in Line 246 of the revised version of the manuscript.
Point 2: In Figure A3, some texts were not shown correctly.
Response 2: Thank you for this point. The alignment of the figures in Figure A3 is improved in the revised manuscript and all the text is clearly readable now.
Reviewer 2 Report
Summary:
They propose a method to identify layers in common CNNs (Inception, ResNet) trained on ImageNet that have highest covariance (=lowest invariance) with respect to scaling. They hypothesize and confirm by experiments that for tasks where scale is important pruning at that layer improves performance when transfering the generic network to a specific task. They show for a relevant problem in the field of histopathology the effectiveness of their method by regressing the nuclei area from images with various magnifications.
I find the idea appealing of not just empirically prune the network but to identify at which layers specific properties are learned (here scale invariance) and use this information to optimize transfer learning. However, I think the manuscript needs another iteration of rework, as in current state it is hard to get the „big picture“ and to grasp the main idea and main hypothesis of the work. Also some parts of the manuscript are redundant and thus it can be shortened (eg. explanation of e^R²/e, magnification categories).
In the following are some suggestions to improve the paper and some questions that arose, grouped by section:
1.Introduction
- Here the main idea or concept, the main hypothesis and main contribution needs to be emphasized much more. Currently it is distributed within row 40 and 65. After reading the introduction for a reader it is not clear what this paper is about and what its contribution is.
Minor:
- I think p-values are not necessary in introduction, the word significant is enough here. In the abstract it is not clear in which context the p-value is used (line 15).
3.Materials and Methods
- Here, I miss a structure. The method itself and the parts necessary for experiments are mixed up. The level of detail changes always, which makes it hard to follow. For a reader also it is often not clear why something was introduced (eg. corrected GAP, white noise images) and if it belongs to the method itself or to a experiment. A stricter separation into method and experiments may help. Also the purpose of a experiment should be described (which hypothesis is it aiming at.) Furthermore, using a top-down approach of providing first an overview to get the main idea of a Section and then break it down to all the details may be usefull.
Minor:
- Why is the normalized R² necessary? Readers familiar with regression should be able to interpret negative R² values, in particular as it is explained in line 200.
4. Results
- I prefer to have experimental setup to be part of the method section or be a separate section and being separated from the quantitative results, as it allows to focus on the results without being distracted of setup details, such as train-test-split. However, this is up the the authors of how to handle the experiments.
- Here, also the purpose of some experiments becomes only clear when reading the discussion (ie. noise inputs). Please clarify either here or in Methods section what purpose a experiment has.
4.4 Improvement of Transfer to Histopathology
- It would be interesting to see the MAE and kappa varies per magnification levels, in particular how it varies between magnification levels.
- A comparison to a baseline model (eg. one from scratch) would be nice, to see whether transfer helps. Also a comparison to a model that works with data augmentation in terms of image scaling to be invariant to magnification would be interesting.
Author Response
Point 1: I find the idea appealing of not just empirically prune the network but to identify at which layers specific properties are learned (here scale invariance) and use this information to optimize transfer learning. I think the manuscript needs another iteration of rework, as in the current state it is hard to get the „big picture“ and to grasp the main idea and main hypothesis of the work.
Response 1: Dear reviewer, thank you for your feedback and the positive comments. Indeed we agree that our approach uses the understanding of inner model mechanics in a proactive way.
We understand your comment. The revised version of the manuscript now presents a clarification of the main assumption and the main hypothesis of this work in lines 32-43:
“This work is based on the assumption that the scale invariance implicitly learned from pretraining on ImageNet can be detrimental to the transfer to applications for which scale is a relevant feature. [...]
The other transferred features such as shape, color and texture, however, are beneficial to the medical tasks, in particular for learning from limited data and improving the models' accuracy and speed of convergence [10, 19-21].
We, therefore, formulate the hypothesis that a specific design retaining helpful features from pretraining while discarding scale invariance would perform better than both a standard transfer and training from scratch.
The experiments in this paper will focus on validating this hypothesis by identifying the network layers where the invariance to scale is learned and by proposing a way to isolate and remove this unwanted behavior while maintaining the beneficial impact of transfer.”
We hope that these important points are now clearer in the paper.
Point 2: Also some parts of the manuscript are redundant and thus it can be shortened (eg. explanation of e^R²/e, magnification categories).
Response 2: Thank you for these suggestions. We have now shortened some parts of the paper. Indeed the magnification categories were repeatedly stated in the manuscript, despite the clear Table 1. We now addressed this redundancy by removing the listing of categories in the text and keeping only Table 1. We have also shortened the explanation of the normalized R^2 in lines 191-193, which has been changed from:
“Note that the $R^2$ normally ranges between zero and one, but it can take negative values.
Different from what one may think, this is not due to a bad choice in the evaluation technique but it rather shows that the prediction on the test samples is far worse than predicting their mean.
To address this, we report the normalized test $R^2$, that is $\frac{e^{R^2}}{e}=e^{R^2-1}$.
In this way, the performance of the RCV on test data is kept in a [0,1] range, with values below $\frac{1}{e}\approx 0.37$ evidencing bad performance. ”
into:
“To keep the test $R^2$ within a [0,1] range, we report the normalized test $R^2=\frac{e^{R^2}}{e}=e^{R^2-1}$, for which values below $\frac{1}{e}\approx 0.37$ evidence bad performance.”.
We removed unnecessary repetitions of the R^2 properties in multiple places in the text. We think that these changes now improved the quality of the revised manuscript.
Point 3: Here the main idea or concept, the main hypothesis and main contribution needs to be emphasized much more. Currently it is distributed within row 40 and 65. After reading the introduction for a reader it is not clear what this paper is about and what its contribution is.
Response 3: Thank you for this comment. Indeed the contributions of the paper and the objectives were not fully clear in the previous version of the manuscript. We now restructured the introduction by stating the context, motivation, and main idea of this work more clearly. The main hypothesis of the paper is now clearly stated in line 32 of the revised manuscript. We also clarified the contributions in lines 53-62 as follows:
“The experiments in this paper extend with new in-depth analyses, results, discussions and visualizations our published research in the Workshop on Interpretability of Machine Intelligence in Medical Image Computing (iMIMIC) at the International Conference on Medical Image Computing and Computer Assisted Intervention (MICCAI2020) [22]. The additional contributions of this paper are stated in the following. New analyses including experiments on image resizing in Section 4.1 and inputs of random noise in Section 4.2 are used to show that object scale and input size have dissociated representations in the CNN layers. While the former is learned from the input data, the latter is shown to be intrinsically captured by the architecture in Section 4.2. The results on the scale quantification are validated for multiple ImageNet object categories in Section 4.3. The significance of the results on the histopathology task is evaluated by statistical testing in Section 4.4. An additional study, besides, is performed on models trained from scratch for this task, showing that our proposed pruning strategy outperforms both these models and pretrained networks in Section 4.4.”
Point 4: Minor: I think p-values are not necessary in introduction, the word significant is enough here. In the abstract it is not clear in which context the p-value is used (line 15).
Response 4: Thank you for this comment. We agree that the context of the statistical testing is not clear in the abstract and introduction, hence we removed, as suggested, the p-values from these sections.
Point 5: Here, I miss a structure. The method itself and the parts necessary for experiments are mixed up. The level of detail changes always, which makes it hard to follow. For a reader also it is often not clear why something was introduced (eg. corrected GAP, white noise images) and if it belongs to the method itself or to an experiment. A stricter separation into method and experiments may help. Also the purpose of an experiment should be described (which hypothesis is it aiming at.) Furthermore, using a top-down approach of providing first an overview to get the main idea of a Section and then break it down to all the details may be useful.
Response 5: This is an important comment. Indeed, parts of the methods section were not clear and needed restructuring and simplification. To address the remarks in this comment, we started by restructuring the section and applying a top-down approach, as suggested, that gives an overview of the Section. The revised version of the manuscript starts now Section 3 with a description of its different parts (lines 115-120):
“This section outlines the proposed method and the setups used for the experiments. Section 3.1 introduces the notations in the paper, while Sections 3.2 and 3.3 describe the datasets and network architectures, respectively.
We outline the main approach in Section3.4, while the evaluation metrics are defined in Section 3.5. The hypotheses, scopes and methodologies of the multiple experiments are described in Section 3.6.”
The main method is now separated from the parts necessary for the experiments. Section 3.4 now describes only the proposed approach and a dedicated section (i.e. Section 3.6) was created to describe the experiments. Before introducing each experiment, we clarify its purpose, so that it is clearer why this experiment was performed and why it is important. For example, we now state in Section 3.6 lines 200-202:
“The experiments in Section 4.1 address the main hypothesis that CNNs pretrained on ImageNet are invariant to transformations of the input size. We want to show, in particular, that this behavior is also true for images containing objects that naturally appear at various scales [...] ”
We also clarify in the same section why noise input images and the corrected GAP were used (lines 215-223 and 225-230):
“We formulate the main hypothesis that the scaling operation $g_{\sigma}(\cdot)$ cannot be performed as a simple input reshaping operation because the CNN features encode information about image size differently from object scale. We hypothesize that information about image size is encoded in the features from the padding effect of early convolutional layers. To verify this, we introduce the corrected GAP [...]. [...] If the network encodes information about the image size differently from object scale, then we should be able to regress the size from the noise inputs. Besides, if this information is encoded from the padding at early layers, then the regression with the corrected GAP should fail as this operation discards the edges of the feature maps. ”
The other experiments are now also introduced in a similar way in Section 3.6 of the revised manuscript. Thank you for this suggestion. We hope that these changes make the revised paper easier to follow.
Point 6: Why is the normalized R² necessary? Readers familiar with regression should be able to interpret negative R² values, in particular as it is explained in line 200.
Response 6: Thank you for raising this question, as this point was not clear enough in the original version of the manuscript. Indeed, the R^2 is clearly defined in footnote no. 6. The normalization is necessary to keep the R^2 values limited within a [0,1] range. This allows a more intuitive comparison between the quantities, as an upper and lower bound are given for the comparison. This also helps to keep a good quality of the plots by maintaining values in a fixed range.
As mentioned in Response 2, we clarified this point in the revised version of the manuscript in line 190-192: “To keep the test R^2 within a [0,1] range for visualization and comparison, we report the normalized test R^2[...]”.
If readers may want to directly inspect the non-normalized R^2 values, these are reported without normalization in Table A1.
Point 7: I prefer to have experimental setup to be part of the method section or be a separate section and being separated from the quantitative results, as it allows to focus on the results without being distracted of setup details, such as train-test-split. However, this is up the the authors of how to handle the experiments.
Response 7: Thank you for this point. We agree that the experimental setup needed a separate section, which we have now included in the Methods section (see Section 3.6) in the revised version of the paper. This modification removed many setup details from the Results section, that is now easier to follow.
Point 8: Here, also the purpose of some experiments becomes only clear when reading the discussion (ie. noise inputs). Please clarify either here or in the Methods section what purpose an experiment has.
Response 8: Indeed, some experiments were not clearly introduced in the previous version of the paper. By introducing Section 3.6 on the experimental setups we have now clarified the purpose of each experiment. Lines 222-223, for example, motivate the use of inputs of white noise.
Point 9: It would be interesting to see the MAE and kappa varies per magnification level, in particular how it varies between magnification levels.
Response 9: Thank you for this point. We added these results to Section 4.4, lines 290-293:
“The average MAE (standard deviations reported in brackets) between the true nuclei areas and those predicted by the pruned Inception V3 are respectively 55.33 (31.16) for 5X images, 42.15 (11.39) for 8X, 34.65 (0.15) for 10X, 33.28 (0.69) for 15X, 48.38 (5.26) for 20X and 81.05 (15.67) for 40X images.”
Point 10: A comparison to a baseline model (eg. one from scratch) would be nice, to see whether transfer helps. Also a comparison to a model that works with data augmentation in terms of image scaling to be invariant to magnification would be interesting.
Response 10: We thank you for this important point. The revised version of the manuscript presents more results and the comparison to a baseline model trained from scratch. Our pruning strategy outperforms training from scratch. These new results were added to Table 2 in the revised version of the manuscript.
Some experiments on scale augmentation could be done to obtain invariance to magnification. As our experiments in Section 4.1 already show, ImageNet pretrained networks are actually already invariant to scale. This behavior is opposite to our main objective, which is to retain the scale covariance. As we want to retain the information about scale, we train our model to classify the magnification categories without applying any scaling transformation to the images. Some equivariant network designs may also be used for comparison, but we feel that this is out of the scope of this paper.
Reviewer 3 Report
The paper by Mara Graziani et al discussed the problem of the common practice on pre-training the CNN on natural image database such as ImageNet and suggested method to isolate unwanted invariances in the features deriving from the transfer learning while maintaining the beneficial impact of the pre-training. The researcher first performed experiment to investigate at which layer that the invariance in the neural network is learned and then the researcher proposed a pruning strategy to preserve the scale information in order to reduces the invariance to object scale of the learned features. The performance of the pruning strategy was evaluated using Inception V3 and ResNet50. Studies in this manuscript revealed that the scale information is present at intermediate layer of the neural network and the pruning strategy improves the transfer learning to maintaining the scale-covariance of the feature in histopathology images.
The strengths of this study are:
- The author has a good insight about the problem of the common practice, to pre-train CNN on natural image database, which causes adverse performance for medical imaging tasks.
- The introduction is well justified the needs to perform this study.
- The study design followed a step-by-step approach and the result showed evidence to prove its hypothesis.
However, there are some concerns remain:
- A major concern of this paper is that parts of the content and the result of this paper is already published by the same group of researcher in 2020. The current paper is an extended version of the published research with the same datasets used, some of the wording, pictures and result of the submitted paper is as same as the published paper. Although the author did quote the published paper once at line 50 and mentioned the submitted paper is an extend of their novel approach.
- Graziani, Mara, Lompech, Thomas, Müller, Henning, Depeursinge, Adrien, & Andrearczyk, Vincent. (2020). Interpretable CNN Pruning for Preserving Scale-Covariant Features in Medical Imaging. In Interpretable and Annotation-Efficient Learning for Medical Image Computing (Lecture Notes in Computer Science, pp. 23-32). Cham: Springer International Publishing.
- The result of this study suggested that pruning strategy works for histopathology images with a significant improvement. The use of pruning in CNN was also investigated by Wang et al (2020), the study used an adaptive pruning technique for Transfer Learned Deep Convolutional Neural Network for Classification of Cervical Pap Smear Images, result suggested that DCNN with adaptive pruning had better performance than the DCNN without adaptive pruning.
- Wang, Pin, Wang, Jiaxin, Li, Yongming, Li, Linyu, & Zhang, Hehua. (2020). Adaptive Pruning of Transfer Learned Deep Convolutional Neural Network for Classification of Cervical Pap Smear Images. IEEE Access, 8, 1.
- Another study performed by Fernandes et al. Evaluated the use of network pruning in Chest X-ray images using a generative adversarial neural networks. The result of the study suggested that the pruned network achieved similar performance compared to the original network.
- Fernandes, Francisco Erivaldo, & Yen, Gary G. (2021). Pruning of generative adversarial neural networks for medical imaging diagnostics with evolution strategy. Information Sciences, 558, 91-102.
- The result of this study and the study performed by Wang et al. suggested the pruning strategy would improve the performance of the CNN for histopathological and Cervical Pap Smear Images, whereas the pruning strategy in GAN in Fernandes et al’s study, did not result in performance gain, therefore further studies shall investigate the use of the pruning strategy in different types of neural networks with medical images from different types modalities, such as CT and MRI.
Author Response
Point 1: The introduction is well justified the needs to perform this study. The study design followed a step-by-step approach and the result showed evidence to prove its hypothesis.
A major concern of this paper is that parts of the content and the result of this paper is already published by the same group of researchers in 2020. The current paper is an extended version of the published research with the same datasets used, some of the wording, pictures and result of the submitted paper is as same as the published paper. Although the author did quote the published paper once at line 50 and mentioned the submitted paper is an extended of their novel approach.
Graziani, Mara, Lompech, Thomas, Müller, Henning, Depeursinge, Adrien, & Andrearczyk, Vincent. (2020). Interpretable CNN Pruning for Preserving Scale-Covariant Features in Medical Imaging. In Interpretable and Annotation-Efficient Learning for Medical Image Computing (Lecture Notes in Computer Science, pp. 23-32). Cham: Springer International Publishing.
Response 1: Dear reviewer, thank you for appreciating the relevance of our work. We are glad that you find the description of the problem, the motivation, design and results, the strong points of this study.
Indeed the contributions of this paper were not fully clear and we have now clarified them in the text by making them more explicit. It is true that this paper extends our work in [1]. The extension consists of new analyses, results, discussions and visualizations. The results reported in Figures 5, 6, 8, 9, A2, A3 and Table 2 are so far unpublished and products of new work from our side. The methods in [1] are extended by analyzing the difference between input size and object scale, with new experiments on image resizing and inputs of random noise. In Table 2, besides, we compare our method to transfer learning (as in [1]), but also to training from scratch, with results that were not in [1]. We propose a statistical analysis of these new results, which was also not done in [1].
These new contributions are now stated in the revised version of the manuscript, lines 53 - 62:
“The experiments in this paper extend with new in-depth analyses, results, discussions and visualizations our published research in the Workshop on Interpretability of Machine Intelligence in Medical Image Computing (iMIMIC) at the International Conference on Medical Image Computing and Computer-Assisted Intervention (MICCAI2020) [22]. The additional contributions of this paper are stated in the following. New analyses including experiments on image resizing in Section 4.1 and inputs of random noise in Section 4.2 are used to show that object scale and input size have dissociated representations in the CNN layers. While the former is learned from the input data, the latter is shown to be intrinsically captured by the architecture in Section 4.2. The results on the scale quantification are validated for multiple ImageNet object categories in Section 4.3. The significance of the results on the histopathology task is evaluated by statistical testing in Section 4.4. An additional study, besides, is performed on models trained from scratch for this task, showing that our proposed pruning strategy outperforms both these models and pretrained networks in Section 4.4.”
Point 2: The result of this study suggested that pruning strategy works for histopathology images with a significant improvement. The use of pruning in CNN was also investigated by Wang et al (2020), the study used an adaptive pruning technique for Transfer Learned Deep Convolutional Neural Network for Classification of Cervical Pap Smear Images, result suggested that DCNN with adaptive pruning had better performance than the DCNN without adaptive pruning.
Wang, Pin, Wang, Jiaxin, Li, Yongming, Li, Linyu, & Zhang, Hehua. (2020). Adaptive Pruning of Transfer Learned Deep Convolutional Neural Network for Classification of Cervical Pap Smear Images. IEEE Access, 8, 1.
Response 2: Thank you for this comment. Adding comparisons to other pruning approaches is indeed very relevant and it improves the quality of the manuscript. We are now acknowledging the suggested reference in lines 90 - 105 of the Related Work section:
“Network pruning approaches were proposed in [34,35], with medical applications for PAP smear imaging [36] and Chest X-rays [37]. Pruned networks achieve similar performance, if not better [36], than the original network.”
In the suggested work, the authors propose an adaptive pruning strategy that discards unimportant filters. Filter importance is measured in terms of the average filter and feature map intensities comprehensively. Differently from this work, in our approach, we do not evaluate individual filter importances. An entire layer is considered as a geometric hyperspace, where linear regression can be performed to identify a hyperplane, or a vector, describing the direction of increasing values of object scale. When following this direction, the extracted features are organized in an ascending way of object scale. Our pruning strategy evaluates where in the network such information is lost and discards those layers to preserve the covariance to scale. We clarify now this important point in the Related Work:
“We propose a pruning strategy that, differently from [34,36], focuses on entire layers and that evaluate the layer importance in terms of the scale covariance of the extracted features.”
Point 3: Another study performed by Fernandes et al. Evaluated the use of network pruning in Chest X-ray images using a generative adversarial neural networks. The result of the study suggested that the pruned network achieved similar performance compared to the original network.
Fernandes, Francisco Erivaldo, & Yen, Gary G. (2021). Pruning of generative adversarial neural networks for medical imaging diagnostics with evolution strategy. Information Sciences, 558, 91-102.
Response 3: Thank you for this suggestion. The pruning algorithm in the suggested paper uses evolutionary strategies to gradually eliminate convolutional filters from the generator model in a GAN architecture. This reference has now been added to the Related Work section in line 91:
“Network pruning approaches were proposed [...] and Chest X-rays [37].”
Differently from their work, however, we propose a pruning strategy that is computationally light, much less intense than evolutionary strategies, and that can be applied to any architecture pretrained on ImageNet inputs to improve transfer. We clarify this in the related work to underline the relevance of our contribution:
“Besides, our pruning strategy does not require an explicit design as in [35], nor expensive computations of evolutionary strategies as in [37]. Our method can be applied to any architecture pretrained on ImageNet inputs to understand the scale covariance of intermediate layers and propose a pruning strategy that would improve the transfer to applications where object scale is a relevant feature.”
Point 4: The result of this study and the study performed by Wang et al. suggested the pruning strategy would improve the performance of the CNN for histopathological and Cervical Pap Smear Images, whereas the pruning strategy in GAN in Fernandes et al’s study, did not result in performance gain, therefore further studies shall investigate the use of the pruning strategy in different types of neural networks with medical images from different types modalities, such as CT and MRI.
Response 4: Thank you for this interesting point. Despite not showing performance gains in [37], the pruning strategy still improves the model’s convergence and the possibility of training CNNs on smaller datasets. We clarified this point in the Related Work (lines 93-96) as follows:
“The asset of network pruning is that, even if not providing massive increases in network performance, it improves training convergence and it reduces the number of parameters to be trained, thus the computational complexity of the models [37]. This allows the training and fine-tuning of the models on smaller datasets, as shown by the study on PAP smear images [36].”
We agree that further studies are needed to investigate the impact of applying different pruning strategies to multiple medical applications and modalities. For this reason, we believe that this paper may present important contributions, presenting an application on breast histopathology images that shows both improved performance and reduced complexity. We added this important point to the Discussion in lines 342 - 344:
“The pruned architectures, besides, provide a reduction in complexity, requiring the training of 51 % and 19 % less of parameters respectively for InceptionV3 and ResNet50.”
Round 2
Reviewer 2 Report
The manuscript improved a lot. In particular the changes in Introduction helped to understand what the main aim of the paper is. The Methods/Experiments Section is now easier to follow.
Reviewer 3 Report
Although the author mentioned that the paper is an extend of the previous work, but a large part of the contents is as same as the previous publication.
It is true that additional information is provided in the paper, but these information did not provide a new insight of the pruning technique, so the findings of the two papers are basically the same.